# Heterozygotes Are a Potential New Entity among Homozygotes and Compound Heterozygotes in Congenital Sucrase-Isomaltase Deficiency

**DOI:** 10.3390/nu11102290

**Published:** 2019-09-25

**Authors:** Diab M. Husein, Dalanda Wanes, Lara M. Marten, Klaus-Peter Zimmer, Hassan Y. Naim

**Affiliations:** 1Department of Physiological Chemistry, University of Veterinary Medicine Hannover, Bünteweg 17, 30559 Hannover, Germany; diab.husein@tiho-hannover.de (D.M.H.); dalanda.wanes@tiho-hannover.de (D.W.); 2Department of Pediatrics, University Medical Center Hamburg-Eppendorf, D-20246 Hamburg, Germany; laramarten@gmail.com; 3Department of General Pediatrics and Neonatology, University Medical Center Giessen and Marburg, Feulgenstr, 10-12, D-35392 Giessen, Germany; klaus-peter.zimmer@paediat.med.uni-giessen.de

**Keywords:** congenital sucrase-isomaltase deficiency, intestinal brush border membrane, protein trafficking phenotypes, homozygote, compound heterozygote, heterozygote

## Abstract

Congenital sucrase-isomaltase deficiency (CSID) is an autosomal recessive disorder of carbohydrate maldigestion and malabsorption caused by mutations in the sucrase-isomaltase (SI) gene. SI, together with maltase-glucoamylase (MGAM), belongs to the enzyme family of disaccharidases required for breakdown of α-glycosidic linkages in the small intestine. The effects of homozygote and compound heterozygote inheritance trait of SI mutations in CSID patients have been well described in former studies. Here we propose the inclusion of heterozygote mutation carriers as a new entity in CSID, possibly presenting with milder symptoms. The hypothesis is supported by recent observations of heterozygote mutation carriers among patients suffering from CSID or patients diagnosed with functional gastrointestinal disorders. Recent studies implicate significant phenotypic heterogeneity depending on the character of the mutation and call for more research regarding the correlation of genetics, function at the cellular and molecular level and clinical presentation. The increased importance of SI gene variants in irritable bowel syndrome (IBS) or other functional gastrointestinal disorders FGIDs and their available symptom relief diets like fermentable oligo-, di-, mono-saccharides and polyols FODMAPs suggest that the heterozygote mutants may affect the disease development and treatment.

## 1. Introduction

Digestion of starch, glycogen, sucrose, maltose and other carbohydrates in the intestinal lumen is achieved by the concerted action of a family of microvillar enzymes, the disaccharidases. The digestion of α-glycosidic linkages of carbohydrates commences by salivary and pancreatic α-amylases and is continued in the small intestine by two major mucosal α-glycosidases, sucrase-isomaltase (SI, EC 3.2.148 and 3.2.1.10) and maltase-glucoamylase (MGAM, EC 3.2.1.20 and 3.2.1.3) [1]. The digestive capacities of SI and MGAM cover almost the entire spectrum of carbohydrates that are linked via α-1,2, α-1,4 and α-1,6 linkages and comprise the majority of the typical diet in children and adults. SI exhibits a wide α-glucosidase activity profile and cooperates with maltase (MA) in digesting α-1,4 linkages, the major glycosidic linkages in starchy food. SI accounts in vivo for almost 80% of mucosal MA activity as well as the entire digestive capacity towards sucrose (α-d-glucopyranosyl-(1→2)-β-d-fructofuranoside; SUC) and almost all isomaltase (IM) (1,6-*O*-α-d-glucanohydrolase) activity [2,3]. By virtue of the complementing activities of SI and MGAM the expression of the two enzyme complexes in the mucosa constitutes an absolute requirement for the mucosal digestion of α-d-glucose oligomers that originate from plants (Table 1) [3].

The two enzymes SI and MGAM share striking structural similarities at the protein level and their biosynthetic pathways are also similar. It is unclear though whether they interact with each other and this interaction may modulate each other’s activities. In vitro studies with recombinant forms of the individual subunits of SI and MGAM have proposed a modulation of starch digestion for slow glucose release through “toggling” of activities of mucosal α-glucosidases [4]. This mechanism suggests an interaction between the two enzymes’ complexes that may occur in close proximity to each other. Reduced expression levels or complete absence of intestinal disaccharidases at the cell surface of the enterocytes is associated with carbohydrate maldigestion and malabsorption, most notably described in several cases of genetically-determined sucrase-isomaltase deficiency (CSID) [5,6,7,8,9]. Here, we review the different inheritance forms of CSID and discuss the possible onset of CSID due to heterozygote mutations as a new entity, possibly presenting with milder symptoms.

## 2. Molecular and Cellular Basis of Genetically-Determined Carbohydrate Malabsorption

The main symptom in CSID is osmotic diarrhea, often acidic, since disaccharides can cause an osmotic force, which drives water into the gut lumen [11]. Bloating, stomach pain and gas are further symptoms of CSID. While unequivocal data on the existence of genetically-determined carbohydrate malabsorption due to MGAM do not exist, the maltase and glucoamylase activities of this enzyme complex are substantially reduced in many cases of CSID. One possible explanation for this reduction is that SI contributes to about 60–80% of the total maltose digesting capacity in the intestine. CSID is elicited by single-nucleotide polymorphisms in the coding region of the *SI* gene. These mutations are distributed over both domains of SI [9]. Biochemical, cellular and functional analyses of SI mutants established the concept of phenotypic heterogeneity and classified the SI mutants into groups that vary in their intracellular localization, cell surface localization (apical/basolateral), proteolytic processing and function [9] (Table 2). Some of the SI mutants are blocked in the endoplasmic reticulum (ER) or the ER-Golgi intermediate compartment (ERGIC) and cis-Golgi [8,12] or are normally trafficked along the secretory pathway, but missorted to the basolateral membrane [13,14]. Other mutants undergo aberrant intracellular cleavage [14,15] or are characterized by an increased turnover rate (Figure 1) [9].

## 3. Homozygous and Compound Heterozygous Inheritance in CSID

A decisive factor in the occurrence and severity of CSID is the inheritance form and whether both alleles of the gene are affected by mutations. The first identified mutation in CSID, Q1089P, is in the sucrase domain of SI and elicits retention of SI in the ERGIC and cis-Golgi compartments [19]. Its inheritance is homozygous and is associated with severe symptoms due to a complete absence of sucrase and isomaltase activities and substantial reduction of the maltase activity [17]. Similar severe effects are also elicited by other homozygous mutations, Q117R, L340P, L620P, C635R, that were identified in intestinal biopsy specimens from CSID patients [12,13,17]. Genetic testing of blood samples from a cohort of patients with diagnosed CSID revealed mutations that were identified as compound heterozygotes, for example G1073D, V577G and F1745C [6,7,16]. These mutations together with R1124X belong to the most common mutations in CSID with an estimated frequency of 83% in European descendants. The severity of these mutations stems from the fact that they generate an SI protein phenotype that is intracellularly blocked in the ER [6,7,16]. More recently, several new mutations in the *SI* gene have been tested by genotyping in a panel of patients suffering from irritable bowel syndrome (IBS) symptoms [20,21,22]. Some of the last cited mutations were already found in CSID patients; these findings unraveled a remarkable heterogeneity in the pathogenesis of CSID revealing the unique etiologies of this multifaceted intestinal malabsorption disorder [6,16].

## 4. Heterozygotes in CSID

While homozygote and compound heterozygote inheritance traits are well documented in diagnosed CSID patients, there are also reports of CSID patients with heterozygous genotypes. In addition, recent studies have found an association of CSID-associated heterozygous genotypes with an increased risk for functional gastrointestinal disorders. In these studies, several mutations in the *SI* gene have been identified, such as R774G, C1229Y and G1073D [6,7,16,18]. In theory, heterozygotes should express an SI molecule that is virtually 50% active or transport-competent if one allele is normal. This hypothesis implies that disaccharides can be metabolized to an extent that does not necessarily elicit malabsorption symptoms. Nonetheless, enzymatic levels of sucrase and maltase in some reported heterozygote cases were apparently low, enough to induce symptoms of carbohydrate malabsorption [6,23]. The existence of one mutated allele in CSID suggests that the pathogenesis of CSID depends not only on the biosynthetic, trafficking and functional features of the individual mutants *per se*, but also on the degree of potential regulatory influence of these mutants on wild type SI. Several observations can be provided to explain the potential effect of heterozygote mutations on wild type SI.

### 4.1. The Quaternary Structure of SI

Recent observations have shown that wild type SI dimerizes along the secretory pathway [16]. The interaction between SI monomers occurs likely via the transmembrane domain, as has been shown for several proteins of the medial Golgi in a fashion referred to as kin recognition [24]. Wild type SI protein could be retained intracellularly along the secretory pathway via its direct interaction with an SI mutant that would exhibit a dominant structural hierarchy over the correctly folded wild type [16]. An intact transmembrane domain of a mutant SI would be enough to elicit this dimerization [25]. Most of the mutations in CSID that have been identified to date are in the luminal domains of SI, which implies that the transmembrane domain may remain intact and capable of interacting with wild type SI. This theory is supported by the fact that mutations that lead to truncated forms of SI, such as R1124X and E821X which lack the transmembrane domain and the theoretical ability to interact with the isoforms of wild-type SI, have not been so far reported in a heterozygote background [16].

### 4.2. The SI Biosynthetic Phenotypes in Intestinal Biopsy Specimens Form CSID Heterozygotes

In two cases of CSID, intestinal biopsy specimens were assessed for biosynthesis, glycosylation and processing of SI in an in vitro heterozygote background with the C1229Y or T694P mutation [6]. The resulting biosynthetic phenotypes of SI in the intestinal tissue that contained the normal and the mutated alleles resembled that of the individually expressed SI mutant in a transfected cell model. SI in an intestinal biopsy specimen expressing only the heterozygote C1229Y mutation is partially trafficked between the ER and the Golgi at a 20–25% maturation rate [6]. A similar biosynthetic pattern was also obtained when an SI-C1229Y mutant was expressed in COS-1 cells [7]. Similarly, in biopsy specimens harboring the heterozygous T694P mutation a mannose-rich ER-located protein phenotype was the prevailing form of SI [6] that was also revealed upon individual expression of the mutant SI-T694P in COS-1 cells (unpublished data). Together these studies strongly propose that SI mutants significantly affect the wild type SI protein via a protein–protein mode of interaction along an early secretory pathway. Such an interaction is possible in cases when the transmembrane domain of SI or its mutants is intact conforming thus to the kin recognition model shown for type II membrane glycoproteins.

### 4.3. The Mosaic Structure of SI in the Enterocytes

Another potential explanation for the symptomatic *SI* heterozygote subjects is the mosaic or heterogeneous expression pattern of many disaccharidases in the enterocytes including SI [26,27]. This is perhaps why the normal activity levels of brush border disaccharidases in intestinal biopsy specimens routinely used in gastrointestinal diagnostics vary substantially among individuals with the highest normal levels being more than 3.5-fold that of the lowest normal levels [28]. Regardless of any potential genetic alterations in the *SI* gene that may lead to abnormalities in its structure or function, the gene expression of SI can be downregulated in different regions of the intestinal epithelium that may ultimately be associated with reduced carbohydrate digestion capacity of the intestine [26,27]. Hitherto individuals with *a priori* reduced expression of wild type SI will be more susceptible to develop gastrointestinal symptoms when SI mutants occur in a heterozygote background.

## 5. Future Perspectives

Several studies provide evidence for a multi-factorial etiology of carbohydrate malabsorption disorders, including psychological and physiological factors [29] besides genetic predisposition. The progress made in the last two decades in the genetics of CSID as well as in unraveling basic molecular mechanisms underlying the pathophysiology of CSID has revised initial concepts on the inheritance trait and severity of the disease. The scientific value gained from this knowledge in the etiology of CSID has resulted in better awareness of the disease as well as the development of more reliable diagnostic tools.

While most patients with extremely severe CSID are homozygotes or compound heterozygotes with pathogenic mutations that elicit localization of SI in the ER, milder forms of CSID can be triggered by combinations of less pathogenic mutations (i.e., partially trafficked SI mutants or heterozygotes). Two immediate questions should be addressed that would certainly contribute to a better understanding of the milder forms of CSID.

### 5.1. What Is the Level of the Contribution of the Partner Glycosidase, MGAM, to the Overall Starch-Digestive Capacity and Other Carbohydrates in CSID? 

At present, there is no firm evidence that has precisely assessed the activities of SI or MGAM alone and in combination and compared the levels of activity of one enzyme to the other. These studies are essential in the context of understanding the pathophysiology in CSID, whether the background is heterozygote, compound heterozygote or homozygote. A toggling effect has been described for a potential substrate hydrolysis by recombinant maltase and sucrase in transient transfection systems. Studies using mutants of SI in co-expression systems with MGAM could help explain a potential compensatory role of MGAM in carbohydrate digestion in milder forms of CSID.

### 5.2. Do Mutations in Heterozygotes Elicit CSID-Like Symptoms?

The major question that requires detailed studies at the cellular and molecular levels is whether an interaction between wild type SI and an SI mutant yields an SI protein phenotype that can be considered biochemically pathogenic. Given the interaction between SI molecules along the early secretory pathway, it is reasonable to suggest this type of interaction when transmembrane domains of SI are intact in SI mutants. Here, protein trafficking and activity profiles of SI mutants in a heterozygote background (i.e., wild type SI plus mutant SI) versus wild type SI can be compared to determine the effect of heterozygous mutations on the function and activity of SI. In parallel studies the effects of MGAM in this heterozygous experimental model can be also examined to mimic the in vivo situation and determine whether the attenuated carbohydrate digestion can be restored by MGAM. While all that has preceded suggests that a potential heterozygote entity in CSID may indeed exist, the validity of this concept implies that the symptoms are not due to other SI mutations, for example in the regulatory non-coding regions of the gene or other genes, or the disaccharidase activity is compromised in vitro and/or ex vivo (in biopsy specimens) and finally, the patients felt better upon dietary or recombinant enzyme therapy administration.

### 5.3. Further Clinical and Nutritional Implications of SI Gene Variants

It has been recognized that irritable bowel syndrome (IBS) affects about 10–13% of adults [30,31]. There is evidence that symptoms of a major part of these patients improve under a gluten free as well as a low fermentable oligo-, di-, mono-saccharides and polyols FODMAP diet [32,33] knowing that FODMAPs contain fructans from wheat, rye, barley and oats. The complexity of IBS brings along a subgroup, which does not respond on a FODMAP diet, but expresses SI gene variants with impaired function and may respond to a more causal therapy [18,21]. Dissecting the pathogenesis of IBS is essential to improve its therapeutic approaches. It should be noted that the phenotypic expression and heterogeneity of CSID does not only depend on the structure, (residual) function and epigenetic effects of SI but also on the amount of fed sucrose, motility factors including intestinal transit time and gastric emptying as well as composition of the intestinal microbiome. It is worth mentioning that one of the most frequent conditions in the daily management of a pediatric gastroenterologist, the so called toddler’s diarrhea, is another subgroup of (pediatric) IBS which still needs therapeutic improvement. While the increased absorption capacity for fructose up to ten years of life represents a “natural” compensatory mechanism [34], it would be not surprising if *SI* gene variants, also heterozygotes, are also involved in this condition.

## Figures and Tables

**Figure 1 nutrients-11-02290-f001:**
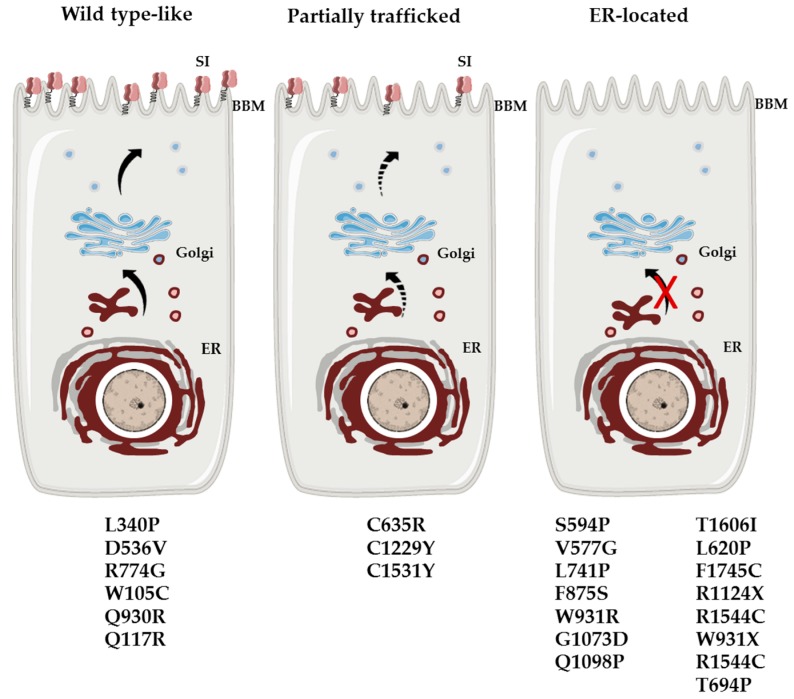
Categorization of the SI mutants into major three biosynthetic protein phenotypes. WT like: the mutants are trafficked along the secretory pathway and mature in a fashion similar to the WT-SI; it is not clear, however, whether an efficient polarized sorting of the mutants to the apical membrane is maintained. Partially trafficked: the mutants are trafficked at a reduced rate between the ER and the Golgi and ultimately to the cell surface. ER block: the mutants are entirely located in the ER. WT: wild type, SI: sucrase-isomaltase, BBM: brush border membrane, ER: endoplasmic reticulum.

**Table 1 nutrients-11-02290-t001:** The amounts of human disaccharidases, sucrase isomaltase (SI), maltase glucoamylase (MGAM) and lactase-phlorizin hydrolase (LPH) in intestinal brush border membrane (BBM) preparation is presented as percentage of total BBM proteins. The activities of the disaccharidases in their natural milieu (BBM) or in immunoprecipitates (Immunopr.) were determined using their respective substrate(s). Adapted from [10].

		SI			MGAM	LPH
Content (% total BBM protein)		8.2 ± 0.7			2.7 ± 1.4	1.4 ± 0.5
Substrate		Sucrose	Isomaltose	Maltose	Maltose	Lactose
Specific activity	Immunopr.	9.5 ± 1.9	5.2 ± 1.4	9.5 ± 1.9	28.1 ± 12.4	1.8 ± 0.3
U·mg^−1^	BBM	27.8 ± 0.5	16.5 ± 0.8	20.2 ± 0.4		5.6 ± 0.3

**Table 2 nutrients-11-02290-t002:** Mutations in congenital sucrase-isomaltase deficiency (CSID), their rs number, allele frequency in the general population (non-Finnish Europeans), position in SI, biosynthetic pattern, activity and CSID genotype.

Amino Acid Change	SNP	Allele Frequency	SI Domain	Biosynthetic Pattern	Activity		CSID Genotype	References
					Sucrase	Isomaltase		
W105C	rs138564183	0.00007054	Isomaltase trefoil 1	Wild type-like	reduced	reduced	Compound heterozygote with W931X	[16]
Q117R	rs121912612	n.d.	Isomaltase	Wild type-like	reduced	reduced	Homozygote	[13]
L340P	rs267607049	n.d.	Isomaltase	Wild type-like	normal	normal	Homozygote	[15]
D536V	rs376816463	0.0002731	Isomaltase	Wild type-like	reduced	inactive	Compound heterozygote with V577G	[16]
V577G	rs121912615	0.002710	Isomaltase	ER-located	inactive	inactive	Compound heterozygote with D536V or G1073D	[6,7,16]
S594P	rs765433197	0.0001091	Isomaltase	ER-located	inactive	inactive	Compound heterozygote with splice site (c.26887 + 1G > C)	[6,16]
L620P	rs121912613	n.d.	Isomaltase	ER-located	inactive	inactive	Homozygote	[12]
C635R	n.d.	n.d.	Isomaltase	Partial trafficked	reduced	reduced	Homozygote	[17]
T694P	n.d.	n.d.	Isomaltase	n.d	n.d	n.d	Heterozygote	[6]
L741P	rs1167931116	0.00006486	Isomaltase	ER-located	inactive	inactive	Compound heterozygote with F1745C	[16]
R774G	rs147207752	0.001401	Isomaltase	Wild type-like	reduced	reduced	Heterozygote	[18]
F875S	n.d.	n.d.	Isomaltase	ER-located	inactive	inactive	Heterozygote	Unpublished
Q930R	rs150927256	0.0004030	Isomaltase	Wild type-like	normal	normal	Compound heterozygote with R1544C	[16]
W931R	rs914403158	0.000008899	Isomaltase	ER-located	reduced	reduced	Compound heterozygote with T1606I	[16]
W931X	rs1314243578	n.d.	Isomaltase	ER-located	inactive	inactive	Compound heterozygote with W105C	[16]
G1073D	rs121912616	0.002313	Sucrase	ER-located	inactive	inactive	Compound heterozygote with R1544C, D577 or R1124X/Heterozygote	[6,7,16]
Q1098P	rs121912611	0.00003882	Sucrase	ER-located/cis-Golgi	inactive	inactive	Homozygote	[19]
R1124X	rs200451408	0.0001629	Sucrase	ER-located	inactive	inactive	Compound heterozygote with G1073D	[16]
C1229Y	rs121912614	0.000008825	Sucrase	Partial trafficked	inactive	reduced	Compound heterozygote/Heterozygote	[6,7]
R1367G	rs143388292	0.0005431	Sucrase	n.d	n.d	n.d	Compound heterozygote with frame shift (.1648delC)	[6]
C1531Y	n.d.	n.d.	Sucrase	Partial trafficked	inactive	reduced	Compound heterozygote with G1073D	[16]
R1544C	rs1340078396	0.00001776	Sucrase	ER-located	reduced	reduced	Compound heterozygote with Q930R	[16]
T1606I	rs376062850	0.00003900	Sucrase	ER-located	reduced	reduced	Compound heterozygote with W931R	[16]
F1745C	rs79717168	0.001581	Sucrase	ER-located	inactive	inactive	Compound heterozygote with L741P or C1229Y/Heterozygote	[6,7,16]

Abbreviations: ER: endoplasmic reticulum; rs number: clustered RefSNP; n.d.: not determined; SNP: single nucleotide polymorphism.

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
