# Peer review of "Heterozygotes Are a Potential New Entity among Homozygotes and Compound Heterozygotes in Congenital Sucrase-Isomaltase Deficiency"

_nutrients, 2019, doi:10.3390/nu11102290_

Round 1

Reviewer 1 Report

The "Concept paper" by Husein et al is interesting and within the scope of the journal and the special issue, however it may be improved, especially aiming to avoid misconceptions about modes of inheritance, and an overall better clarity of the message the authors are trying to convey

I have the following comments

Major

the terminology is often inaccurate when it comes to modes of inheritance and types of mutations, the authors should re-proof their text avoiding misleading phrasing. for instance, mutations or DNA variants themselves do not show any pattern of inheritance (line 81 "its inheritance trait is homozygous"), rather the trait does. compound inheritance (rarely used terminology, if anything...) is usually referred to conditions where there is a loss-of-function mutation accompanied by a (often common) hypomorphic mutation. there is certainly not enough evidence for such a phenomenon in CSID, so this terminology should be avoided. mutations can still be described as detected in homozygous, heterozygous or compound heterozygous state but this does not imply they (or, better, the trait) can show "compound inheritance" overall, there seems to be not enough experimental or clinical evidence that heterozygosity for SI mutations leads to a medically relevant phenotype ("a new clinical entity"), the authors may want to reconsider their (ambitious) title, while more concept they aim to propose is still valid (although requiring validation) Table 1 suffers from a number of issues, mainly due to ambiguous terminology. First of all, it would be ideal to order the aa changes (called "mutation") according to the progressive residues in the SI protein; these mutations should be flanked by some additional information as to their clinical relevance (eventually including corresponding clinvar entry?), genomic coordinates (rs nr from dbSNP or similar) and frequency in the general population (from gnomad?); "position on SI" should be replaced with SI domain; what does "biosynthetic severity" means...?; what is the difference between "partial" and "reduced" enzymatic activity?; how can a "heterozygote" with normal activity be pathogenic? (unpublished results??); "heterozygote" and "compound heterozygote" refers to individuals not DNA variants, and the heading "inheritance pattern" should be avoided as from previous comment, possibly replaced by the heading "CSID genotype" and groups "compound heterozygous with mutation XXX" etc...; it is also not clear from the table whether these are all the CSID mutations identified so far or a selection (how made?) lines 89-93, this sentence is wrong and misleading, as the quoted work was on IBS and not CSID patients. in fact, also the following lines are inappropriate, since the variants reported in refs 18-19 were NOT tested ("biochemical and cellular investigations of EACH of these mutations...") lines 114-119 and discussion: were the mentioned heterozygous patients also sequenced for the entire SI coding region, could the presence of another CSID mutation be excluded? if not conclusions are speculative and this limitation (lack of seq data) may be acknowledged  it is not clear what the relation is between chapter 3 ("Heterozygotes in functional gastrointestinal disorders") and its subparagraphs a-e...? lines 160-164 about haploinsufficiency, once again these are misleading as there is no such unequivocal evidence, neither it was provided in the quoted ref (17) lines 167-168, how can intronic deletions/mutations lead to frame shift and stop codons?? (in exon-sequenced individuals) lines 171-172, this estimation is completely wrong and unsubstantiated, as it was wrongly quoted in ref 26, which in turn only quotes an abstract where just a familial case is reported and everything else is pure speculation although the focus of their review is on SI, the authors often refer to MGAM throughout the paper. it would be ideal to have a figure included, where the relative expression in the GI tract, as well as the specific disaccharidase activities and substrates are reported for these two enzymes

Minor

ref 18 is the same as ref 31, while a key ref appears to be missing (PMID: 27872184), possibly meant to be ref 18? line 60 "... since molecules (especially those of smaller size)..." what molecules? the terminology may be ambiguous at times, it is advisable to refer to mutations for CSID-causing DNA variants (as from detection in CSID patients) and SNPs for DNA variants of less certain causative role. ie like those detected in heterozygous CSID and IBS patients (since these are also detected in controls) figure 1, referring to heterozygous, homozygous and compound heterozygous genotype is again misleading and should be avoided lines 108-109 are inaccurate, in the quoted studies (ie ref 17) the authors did not identify mutations in the SI gene, but tested rare variants (by genotyping) as these were predicted to correspond to deleterious changes (still, no experimental data on their relative enzymatic activity)

Author Response

Reviewer 1

We thank the reviewer for the suggestions and constructive comments, which have been addressed in the revised version of the manuscript. All changes made are marked.

We have dissected the comments raised by this reviewer and in what follow we respond to each of the issues raised.

Major

The terminology is often inaccurate when it comes to modes of inheritance and types of mutations, the authors should re-proof their text avoiding misleading phrasing. for instance, mutations or DNA variants themselves do not show any pattern of inheritance (line 81 "its inheritance trait is homozygous"), rather the trait does, Compound inheritance (rarely used terminology, if anything...) is usually referred to conditions where there is a loss-of-function mutation accompanied by a (often common) hypomorphic mutation. there is certainly not enough evidence for such a phenomenon in CSID, so this terminology should be avoided.

Response:

Oftentimes the frequent use of laboratory jargon leads to unintentional mix of terms that ends up with some kind of inaccuracy for the non-intended reader. We therefore followed the comments raised by this reviewer and made changes where appropriate.

 In fact, the mutations can still be described as detected in homozygous, heterozygous or compound heterozygous state but this does not imply they (or, better, the trait) can show "compound inheritance" overall, there seems to be not enough experimental or clinical evidence that heterozygosity for SI mutations leads to a medically relevant phenotype ("a new clinical entity"), the authors may want to reconsider their (ambitious) title, while more concept they aim to propose is still valid (although requiring validation)

Response:

In a study with 11 patients suffering from diarrhea, five mutations were identified to exist as heterozygotes (Sander et al., 2006). The activities in these cases were substantially reduced, but not entirely absent, which is likely due to the existence of one unaffected allele of SI. Another study comprising 9 cases, 7 have been identified as compound heterozygotes and have been published (Gericke et al., 2017) and 2 heterozygotes reported in this manuscript as unpublished. The R774G has been recently published (Husein and Naim, 2019) (this has been now corrected in the Table) and the second one F875S is entirely inactive. The effects of these heterozygote mutations on the overall activities of SI in biopsy samples from these patients were substantial (R774G) or have led to complete abolishment of the SI activities (F875S) (included in the Table).

The clinical phenotypes as assessed by the enzymatic activities have been published before and are cited in the Table where appropriate.

In view of all this, we think that heterozygotes are a new entity in SI deficiency. We removed clinical from the title.

 Table 1 suffers from a number of issues, mainly due to ambiguous terminology. First of all, it would be ideal to order the aa changes (called "mutation") according to the progressive residues in the SI protein; these mutations should be flanked by some additional information as to their clinical relevance (eventually including corresponding clinvar entry?), genomic coordinates (rs nr from dbSNP or similar) and frequency in the general population (from gnomad?); "position on SI" should be replaced with SI domain; what does "biosynthetic severity" means...?; what is the difference between "partial" and "reduced" enzymatic activity?; how can a "heterozygote" with normal activity be pathogenic? (unpublished results??).

Response:

We agree that Table 1 contains several unclarities and we have modified it following most of the comments raised by the reviewer, made a few corrections and hope it is now more informative. We changed "partial" to "reduced" throughout. There has been an error in presenting the R774G mutant, which has now been corrected. The trafficking of this mutant is normal, like wild type, but the activities are reduced as has been shown before (Husein and Naim, 2019). We added rs Nr and frequency in the general population.

 "heterozygote" and "compound heterozygote" refers to individuals not DNA variants, and the heading "inheritance pattern" should be avoided as from previous comment, possibly replaced by the heading "CSID genotype" and groups "compound heterozygous with mutation XXX" etc...; it is also not clear from the table whether these are all the CSID mutations identified so far or a selection (how made?)

Response:

These are the mutations that have been analyzed in our laboratory and are by far not all SI mutations. 

 lines 89-93, this sentence is wrong and misleading, as the quoted work was on IBS and not CSID patients. in fact, also the following lines are inappropriate, since the variants reported in refs 18-19 were NOT tested ("biochemical and cellular investigations of EACH of these mutations...")

Response:

The sentence is now corrected; some of the ´IBS´ mutants were already analyzed in CSID patients (e.g. R1124X and R1367G) and the references are now cited (Sander et al., 2006; Gericke et al., 2017; Garcia-Etxebarria et al., 2018).

 lines 114-119 and discussion: were the mentioned heterozygous patients also sequenced for the entire SI coding region, could the presence of another CSID mutation be excluded? if not conclusions are speculative and this limitation (lack of seq data) may be acknowledged 

Response:

The entire coding region has been sequenced (Sander et al., 2006; Gericke et al., 2017).

 It is not clear what the relation is between chapter 3 ("Heterozygotes in functional gastrointestinal disorders") and its subparagraphs a-e...?

Response:

The reviewer means point #4 and not #3. In any case, the title has been changed to "Heterozygous trait of inheritance in CSID".

 lines 160-164 about haploinsufficiency, once again these are misleading as there is no such unequivocal evidence, neither it was provided in the quoted ref (17)

Response:

The correct reference is now provided that describes the identification of mutations on one single allele (Sander et al., 2006).

 lines 167-168, how can intronic deletions/mutations lead to frame shift and stop codons?? (in exon-sequenced individuals)

Response:

This sentence was corrected.

 lines 171-172, this estimation is completely wrong and unsubstantiated, as it was wrongly quoted in ref 26, which in turn only quotes an abstract where just a familial case is reported and everything else is pure speculation

Response:

The cited paper is already published, but because of the confusion, the sentence was deleted.

 although the focus of their review is on SI, the authors often refer to MGAM throughout the paper.

Response:

As indicated in the "Introduction", SI exhibits a wide α-glucosidase activity profile and cooperates with maltase-glucoamylase (MGAM) in digesting α-1,4 linkages, the major glycosidic linkages in starchy food. As a matter of fact, SI accounts in vivo for almost 80% of mucosal MGAM activity as well as the entire digestive capacity towards sucrose. Obviously pathogenic mutants of SI have also implications on the overall starch digestion and we strongly believe that this should be mentioned in a paper on SI mutants.

 it would be ideal to have a figure included, where the relative expression in the GI tract, as well as the specific disaccharidase activities and substrates are reported for these two enzymes

Response:

We refer the reviewer to a publication in Nutrients that describes these issues (Amiri and Naim, Characterization of Mucosal Disaccharidases from Human Intestine

Nutrients 2017, 9(10), 1106; https://doi.org/10.3390/nu9101106).

 Minor

ref 18 is the same as ref 31, while a key ref appears to be missing (PMID: 27872184), possibly meant to be ref 18?

Response:

The reference has been changed.

 line 60 "... since molecules (especially those of smaller size)..." what molecules?

Response:

This is now changed "…since dissacharides can cause an osmotic force…."

 the terminology may be ambiguous at times, it is advisable to refer to mutations for CSID-causing DNA variants (as from detection in CSID patients) and SNPs for DNA variants of less certain causative role. ie like those detected in heterozygous CSID and IBS patients (since these are also detected in controls)

Response:

The CSID mutations that we have characterized have been classified into three categories based on the trafficking behavior of the SI mutants between the ER, Golgi and cell surface (Gericke et al., 2017). Further, in view of the variations in the enzymatic activities among the mutants we proposed the existence of several forms of CSID that vary in their severity (severe-intermediate-mild). Moreover, some of the CSID mutations are also identified in IBS. It will be therefore confusing for the readers to change the terminology in the Table. 

figure 1, referring to heterozygous, homozygous and compound heterozygous genotype is again misleading and should be avoided

Response:

This is now changed.

lines 108-109 are inaccurate, in the quoted studies (ie ref 17) the authors did not identify mutations in the SI gene, but tested rare variants (by genotyping) as these were predicted to correspond to deleterious changes (still, no experimental data on their relative enzymatic activity

Response:

This comment is not correct. In fact, reference 17 (Gericke et al., BBA 2017) describes “A total of 31 probands with a confirmed biopsy diagnosis of CSID” (section 2.3). This implies that the enzymatic activities in the biopsies were examined and found to be reduced or absent. Also other mutations described in this manuscript have been identified after clinical diagnosis with CSID (several references are provided in this manuscript). DNA sequencing of the entire coding region of the SI gene was performed and the mutations identified. We have analyzed these mutations at the biochemical and functional levels.   

Reviewer 2 Report

Husein and colleagues present a concept piece focused on the possibility that genetic variants That impact on sucrose digestion could cause functional gastrointestinal symptoms and other issues.

The concept is reasonable and it may be that mutations in genes involved in sucrose digestion explain functional gastrointestinal symptoms in some subjects. This is supported by case reports of various mutations in patients; however, case-controlled studies are not available.

The authors should discuss clinical prsentation of homo- and hetero-zygote manifestations in more detail (including age of patient). Are acquired forms also described?

The clinical expression of carbohydrate malabsorption depends not only on the genetic predisposition but also other physiological and psychological factors (see Deng etal Nutrients review on Lactose Digestion 2015) This should be mentioned. 

Given the wide variety in potential functionally relevant mutations, a bioassay of small bowel sucrase activity/ digestion may be more practical than deep sequencing. Work is needed to establish contribution of these genetic variations to carbohydrate malabsorption in clinical practice.

Certain technical sections are very densely written and could benefit from editing and figures to ilustrate mechanism

Author Response

Reviewer 2

We thank the reviewer for the suggestions and constructive comments, which have been addressed in the revised version of the manuscript. All changes made are marked.

The authors should discuss clinical presentation of homo- and hetero-zygote manifestations in more detail (including age of patient). Are acquired forms also described?

Response:

Several studies on CSID that are cited in this paper have presented details regarding the clinical manifestations as well as the age of the patients. In other studies (Gericke et al., BBA 2017) we did not receive information on the age of the patients.

The clinical expression of carbohydrate malabsorption depends not only on the genetic predisposition but also other physiological and psychological factors (see Deng et al Nutrients review on Lactose Digestion 2015) This should be mentioned.

Response:

The comment is well taken and the reference is added (line 172).

Given the wide variety in potential functionally relevant mutations, a bioassay of small bowel sucrase activity/ digestion may be more practical than deep sequencing. Work is needed to establish contribution of these genetic variations to carbohydrate malabsorption in clinical practice.

Response:

The mutants described in this paper were first analyzed at the clinical level in biopsy specimens followed by genetic testing and exome sequencing.  Certain technical sections are very densely written and could benefit from editing and figures to illustrate mechanism

Response:

We are not certain, which sections the reviewer refers to. We think that the most important mechanistic aspect in this paper is provided by Figure 1 that schematically summarizes the trafficking pattern of the SI mutants and their description in the text is required to discriminate between the different SI mutants.

Round 2

Reviewer 1 Report

Unfortunately, several issues still remain, and I therefore have mostly comments similar to those already expressed at the first submission stage, namely again:

1) The terminology is often inaccurate, neither heterozygous trait, or pattern or else is appropriate. If the authors want to propose SI mutant heterozygosity as a clinical entity (highly speculative), this does not imply anything when it comes to the pattern of inheritance, (ie there is no such heterozygous inheritance), it simply means that individuals with a single "mutant" copy of the SI gene may show milder but still clinically relevant symptoms, compared to homozygous or compound heterozygous individuals. The whole manuscript should be (extensively) revised accordingly.

2) looking at the table, there dont seem to be so many cases of heterozygotes showing a clinical CSID or CSID-like phenotype. In order to propose such "entity" one would need to be sure that the symptoms are not due to 1) other SI mutations, for example in the regulatory non-coding regions of the gene (not done) or b) other genes, c) that disaccharidase activity is compromised in vitro and/or ex vivo (biopsies) and eventually d) that the patients got better upon dietary or recombinant enzyme therapy administration. These considerations must be made and discussed at a minimum in the context of such a paper, including the fact that most of the mutations occur in the general population (although at very low frequency)

3) the authors may want to reconsider their (ambitious) title, I could not see any changes in the revised version. 

4) it may not be appropriate to list unpublished data and unpublished patients characterization in support of the hypothesis, when this is primarily a concept/review paper (ie the actual results are not reported)

5) Table 1 still has problems, ie information is still missing for several SNPs (rs nr) the population should be specified (should be non-finnish europeans) and it is not clear what the heading is for the frequency, neither what the marks mean # etc

6) lines 95, "identification of several new mutations in the SI gene", again this is misleading, as these mutations were not identified in the study, they were tested by genotyping

7) It is still not clear AT ALL what the relation is between chapter 4 ("Heterozygotes in functionsal gastrointestinal disorders") and its subparagraphs a-e...? how are these paragraphs related to the inheritance? paragraphs a-c certainly belong somewhere else, and again there are misleading concepts (including haploinsufficiency, unproven), the authors need to revise their statements about inheritance throughout the paper

8) lines 167-168, still the same problem after revision, how can intronic deletions/mutations lead to frame shift and stop codons?? 

9) I still believe it would be ideal to have a figure included, where the relative expression in the GI tract, as well as the specific disaccharidase activities and substrates are reported for these two enzymes

10) PMID: 27872184 is still missing, this is the original study where SI mutations were described in IBS

Author Response

We thank the reviewer for the stimulating and thorough review of this manuscript. Below are our responses to the specific points raised and the changes are marked in the revised manuscript.

The terminology is often inaccurate, neither heterozygous trait, or pattern or else is appropriate. If the authors want to propose SI mutant heterozygosity as a clinical entity (highly speculative), this does not imply anything when it comes to the pattern of inheritance, (ie there is no such heterozygous inheritance), it simply means that individuals with a single "mutant" copy of the SI gene may show milder but still clinically relevant symptoms, compared to homozygous or compound heterozygous individuals. The whole manuscript should be (extensively) revised accordingly.

Response:

We have done our best to remove whatever is related to ”heterozygous trait or pattern”. If still needed, we would be happy to include any constructive suggestions by the reviewer to improve the text at this stage.

looking at the table, there dont seem to be so many cases of heterozygotes showing a clinical CSID or CSID-like phenotype. In order to propose such "entity" one would need to be sure that the symptoms are not due to 1) other SI mutations, for example in the regulatory non-coding regions of the gene (not done) or b) other genes, c) that disaccharidase activity is compromised in vitro and/or ex vivo (biopsies) and eventually d) that the patients got better upon dietary or recombinant enzyme therapy administration. These considerations must be made and discussed at a minimum in the context of such a paper, including the fact that most of the mutations occur in the general population (although at very low frequency).

Response:

We took these suggestions into consideration and added a paragraph to allude to these issues (5.Future perspectives, under b.). The paragraph is based almost entirely on the reviewer’s comments. 

the authors may want to reconsider their (ambitious) title, I could not see any changes in the revised version. 

Response:

We have removed clinical from the initial title in the first revision. Now we have added “potential” to soften the title. We would be happy to consider suggestions by the reviewer for another suitable title if needed.

it may not be appropriate to list unpublished data and unpublished patients characterization in support of the hypothesis, when this is primarily a concept/review paper (ie the actual results are not reported)

Response:

The only unpublished data are those that belong to the mutants T694P and R1367G. We do have the entire data sets for these 2 mutants, but we thought it is not appropriate to make a mix of biochemical/cell biology data of two mutants within a concept paper.

Table 1 still has problems, ie information is still missing for several SNPs (rs nr) the population should be specified (should be non-finnish europeans) and it is not clear what the heading is for the frequency, neither what the marks mean # etc

Response:

The missing information (rs nr or frequencies) were not determined. The allele frequency in general European population (non-Finnish) was corrected.

lines 95, "identification of several new mutations in the SI gene", again this is misleading, as these mutations were not identified in the study, they were tested by genotyping

Response:

We agree. This sentence has now been changed.

It is still not clear AT ALL what the relation is between chapter 4 ("Heterozygotes in functionsal gastrointestinal disorders") and its subparagraphs a-e...? how are these paragraphs related to the inheritance? paragraphs a-c certainly belong somewhere else, and again there are misleading concepts (including haploinsufficiency, unproven), the authors need to revise their statements about inheritance throughout the paper

Response:

We have actually changed in the first revision the title of chapter 4. Now it is changed again to read: 4. Heterozygotes in CSID. We have also completely deleted the two points d and e in this chapter (haploinsufficiency and the intronic variations).

lines 167-168, still the same problem after revision, how can intronic deletions/mutations lead to frame shift and stop codons?? 

Response:

This is deleted in this version.

I still believe it would be ideal to have a figure included, where the relative expression in the GI tract, as well as the specific disaccharidase activities and substrates are reported for these two enzymes

Response:

A table with the required information is now included that is adapted from a previous paper by Amiri and Naim (Nutrients 2017).

 10) PMID: 27872184 is still missing, this is the original study where SI mutations were described in IBS

Response:

The mentioned reference is now cited (Ref. 19).